# Factors associated with unmet healthcare needs in patients using Primary Care Access Points for unattached patients in Quebec (Canada)

Mylaine Breton[1]*, Catherine Lamoureux-Lamarche[1], Véronique Deslauriers[1], Djamal Berbiche[1], Maude Laberge[2,3,4], Annie Talbot[5], Aude Motulsky[6], Marie-Pascale Pomey[6], Isabelle Gaboury[1]

1 Centre de Recherche Charles-Le Moyne, Université de Sherbrooke, Campus Longueuil, Longueuil, Canada, 2 Département de Médecine Sociale et Préventive, Faculté de Médecine, Université Laval, Québec, Québec, Canada, 3 VITAM, Centre de recherche en santé durable, Université Laval, Québec, Québec, Canada, 4 Centre de recherche du CHU de Québec, Université Laval, Québec, Québec, Canada, 5 Département de médecine de famille et de médecine d'urgence, Faculté de Médecine, Université de Montréal, Montréal, Québec, Canada, 6 Centre de Recherche du Centre Hospitalier de l'Université de Montréal, School of Public Health, Université de Montréal, Montréal, Canada

* mylaine.breton@usherbrooke.ca

## Abstract

### Background

Access to primary care is an important component of health systems. Given the barriers experienced by unattached patients to accessing primary care in Quebec (Canada), the Ministry of Health mandated the province-wide implementation of Primary care access points for unattached patients (*Guichet d'accès première ligne*; GAP), an organizational innovation designed to orient patients to the most appropriate professional or service. This study aims to 1) document the factors associated with unmet healthcare needs after receiving GAP services and 2) assess whether those factors vary by GAP orientation.

### Methods

This cross-sectional study builds on data collected between April and July 2024 using an online patient questionnaire. All patients with a valid email address registered on the centralized waiting list for unattached patients in three local health territories (LHTs) received an email invitation to participate in the survey. The total sample included 20,282 participants who responded to the questionnaire and used the GAP.

### Results

The findings showed that younger age, self-reporting poor/fair physical and mental health, receiving services in LHT 3 and reporting an emergency room visit were associated with increased likelihood of reporting unmet needs. Stratified analyses

which permits unrestricted use, distribution, and reproduction in any medium, provided the original author and source are credited.

**Data availability statement:** Data cannot be shared publicly because of confidentiality of information. Participants did not give consent to share their data. Requests should be made to the Research Ethics Committee of the Centre intégré de santé et de services sociaux Montérégie-Centre (cr-info.cisssmc16@ssss.gouv.qc.ca).

**Funding:** The study was funded by a Catalyst Grant #475314 (MB, ML) awarded by the Canadian Institutes of Health Research (https://cihr-irsc.gc.ca/e/193.html) and a Grant #5-2-01 (MB) from the Fonds de Soutien à l'innovation en santé et services sociaux (https://www.medteq.ca/en/). The funders did not play any role in the study design, data collection and analysis, decision to publish, or preparation of the manuscript.

**Competing interests:** I have read the journal's policy and the authors of this manuscript have the following competing interests: AT mentioned that a family member is working for a pharmaceutical company. She also received an honorarium as consultant to evaluate the GAP implementation in Quebec from MSSS.

suggested that some characteristics (age, use of emergency room) were associated with unmet needs across orientations, while others (self-reported physical and mental health) were associated with specific orientations.

## Conclusion

This study serves as a first step in deepening our understanding from a patient perspective of how to better plan primary care services and improve unattached patients' experiences using the GAP. The findings showed that patients oriented to other professionals than a medical appointment with a family physician had the highest percentage of unmet needs. The next step involves an in-depth exploration of the reasons for patients' unmet needs, enabling the development of more precise and effective strategies to address them.

## Introduction

Primary care is the entry point to the healthcare system [1–3], where 80% of patients' healthcare needs are met throughout their life course [4]. Access to primary care has, however, been a major issue in Canada for the last 20 years, leading to a primary care crisis [5,6]. In 2022, it was estimated that 6.5 million (22%) Canadians were unattached to a regular primary care provider (PCP), and the province of Quebec was among the worst with 31% of unattached patients [7,8]. The benefits of being attached to a PCP are well documented, including better preventive and chronic care, improved health outcomes and fewer emergency room visits and avoidable inpatient stays [9–12]. Unattached patients are also more likely to report unmet healthcare needs [13–15].

To address access challenges, seven Canadian provinces including Quebec have implemented centralized waiting lists (CWL) for unattached patients, where all requests for attachment are centralized [16,17]. Patients and PCP are geographically matched based on PCP capacity as well as priority criteria in some provinces [16]. Given the magnitude of access challenges in Quebec, including the long waiting periods prior to attachment through the CWL, the Ministry of Health and Social Services mandated, in 2022, the implementation of Primary care access points for unattached patients (*Guichet d'accès à la première ligne*; GAP) across the province. This organizational innovation serves as an entry-point to primary care for unattached patients while they are awaiting attachment on the CWL. When patients encounter a prompt healthcare need and contact the GAP, they are evaluated and oriented towards the most appropriate health professional or service. The GAP aims to make optimal use of the expertise of healthcare professionals other than family physicians to enhance timely access to care. Against a backdrop of shortages in the healthcare sector and an increased numbers of family physicians retiring, appropriate orientation of patients is paramount. As of November 2024, more than 1.6 million individuals were registered on the CWL across the province and were eligible to use the GAP [18].

Given the novelty of the GAP, few studies have documented the characteristics of its users or their perceptions of care. To our knowledge, no previous study has collected patient-reported experience measures (PREM) related to the GAP, including unmet healthcare needs. Unmet healthcare needs are defined as "*perceived needs for receiving healthcare services that are not obtained*" [15] and are associated with deteriorating health [19,20]. Unmet healthcare needs are an important indicator in the evaluation of healthcare systems, particularly in access to care [21,22]. Assessing unmet healthcare needs may enable the identification of areas for improvement in the GAP clinical orientations where patients' expectations are not met in terms of accessibility, availability, acceptability and quality [22].

The factors associated with unmet healthcare needs have been widely documented in Canada and include younger age, female gender, higher education, being separated or divorced, having lower income and reporting poorer health status and chronic diseases [13–15,23–28]. The factors documented in other countries with a universal healthcare system are similar, but the status of employment (unemployed) is frequently associated with unmet healthcare needs [29–31], whereas results are mixed in Canadian studies. However, studies focused specifically on primary care or unattached patients are limited. Therefore, the objectives of this study are to 1) document the factors associated with unmet healthcare needs after receiving a GAP service and 2) assess if these factors vary according to GAP service received. Given that the GAP aims to facilitate access to care for unattached patients, notably by using the expertise of health professionals other than family physicians, assessing whether orientations other than an appointment with a family physician meet patients' needs will help establish the "proof of concept" of implementing GAPs province-wide.

## Materials and methods

### Primary care access points (Guichet d'accès à la première ligne, GAP)

A visual representation of the GAP is presented in Fig 1. Each GAP is linked to the local CWL and registration on the CWL is required to use the GAP. Patients with healthcare needs filled an online form–digital GAP– by themselves or called a central number where the form is completed by an administrative clerk. At the end of the form, the patient can receive one or more of the following: self-care advice, orientation to specific trajectories (e.g., appointment for vaccination) and/or instruction to wait to be called back by the GAP. If required, patients' requests are categorized according to a priority scale, and wait times to be contacted and assessed by GAP staff vary accordingly. Then, they may be oriented to the following professionals or services according to the clinical need and available resources: medical appointment with a family physician or nurse, reference to another health professional (e.g., physiotherapist, psychologist) or community pharmacist, orientation to the emergency room or other service (e.g., community organization, local community health center). Family physicians received incentives to offer time slots for GAP patients.

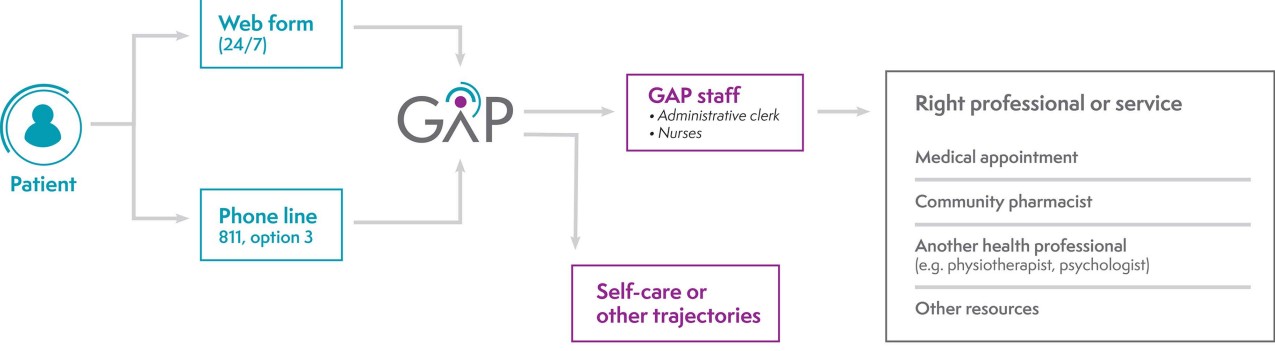

**Fig 1. GAP processes and orientations.**

## Design and setting

This cross-sectional study is part of a larger longitudinal mixed-methods case study that aims to analyze the implementation of Primary care access points for unattached patients [32]. This study took place in the Montérégie region of Quebec (Canada), the second most populous region of the province, which includes nearly 18% of Quebec's population and more than 297,000 unattached patients registered on the CWL. The region is divided into three local health territories (LHTs) each with one GAP. The characteristics of the three LHTs are presented in Table 1. The study was approved by the Research Ethics Committee of the Centre intégré de santé et de services sociaux Montérégie-Centre (MP-04-2023-716) and all participants gave informed electronic consent, which consisted of clicking the box ''I agree to participate in this research project" at the end of the consent form. We follow the Strengthening the reporting of observational studies in epidemiology (STROBE) guidelines to report our study [33].

## Data collection

Data were collected using an online patient questionnaire developed based on both pre-existing health questionnaires [36–50] and our expertise in GAP design, as no instrument existed to document patient experiences of the GAP. It was composed of four sections: 1) experience navigating the health system as an unattached patient, 2) health service utilization, 3) GAP service experience and 4) socio-demographic, economic and clinical characteristics. The questionnaire included 72 questions.

The development of the questionnaire involved several phases, incorporating feedback from researchers, GAP managers and patients, as well as cognitive testing [51] conducted in a community organization with users to identify potential challenges to completing the questionnaire for patients with low health literacy. Fifteen patients participated in cognitive testing of the questionnaire. Following these tests, the wording of some questions was revised.

The questionnaire was administered between April 16th and July 10th, 2024. All patients with a valid email address registered on the CWL for unattached patients in the three LHTs under study received an email invitation to participate. A reminder email was sent after 7 days to all patients who had not opened the questionnaire. The email reminder approximately doubled the number of participants. Of the 279,000 unattached patients eligible for the three GAPs, an invitation to participate was sent to the 212,546 patients who had a valid email. A total of 41 384 individuals (19.47%) responded to the questionnaire. Participants were allowed to answer for themselves, their children or an adult for whom they were directly

**Table 1. Characteristics of the three LHTs included in the study [18,34,35].**

|  | LHT 1 | LHT 2 | LHT 3 |
|---|---|---|---|
| Population (2024)[a] | 510,638 | 557,115 | 444,951 |
| Patients registered on the CWL[b] | 69,303 | 113,308 | 90,856 |
| Number of GAP requests (7-day moving average)[b] | 183 | 275 | 263 |
| Average time to complete GAP request[b] | 124h 35 min | 166h 29 min | 60h 14 min |
| Orientation through GAP[b] |  |  |  |
| Medical appointment with a family physician[b] | 48.50% | 48.60% | 57.00% |
| Appointment not available | 6.50% | 4.30% | 5.20% |
| Community pharmacist | 3.00% | 4.10% | 4.80% |
| Other | 28.40% | 24.60% | 17.40% |
| No referral required | 13.60% | 18.40% | 15.60% |

CWL: Centralized Waiting List for unattached patients; GAP: Primary Care Access Point for Unattached Patients (*Guichet d'accès à la première ligne*); LHT: Local Health Territory

[a]Projections for 2024

[b]Statistics between April and May 2024

involved in their care. However, the section on socio-demographic, economic and clinical characteristics was addressed to the respondent. Given that the factors (independent variables) considered in this study came mostly from this section, the 37,685 individuals who responded for themselves were included in the study sample. Of those, 20,282 individuals (53.82%) had used the GAP and were included in the study sample.

## Measures

The Andersen behavioral model of health services utilization [52] was used to identify predisposing, enabling and need factors potentially associated with perceived unmet healthcare needs related to GAP services received. This model has been widely used in previous studies on unmet healthcare needs and access to healthcare services [14,53–55].

**Dependent variable.** All patients who used the GAP were asked: *"For the main reason you called the GAP, which professional or service did the GAP refer you to for your health needs?".* Respondent were asked to select the main professional or service based on the primary reason for call. Unmet healthcare needs measured were specific to this main professional seen or service received using the following question: *Did this [professional, service or information received] meet your health needs?* (yes/no).

**Independent variables. Predisposing factors** included gender (male/female), age group (18–34; 35–54; 55–69; ≥ 70) and being born in Canada (yes/no).

**Enabling factors** included the social vulnerability index, GAP orientation and the LHT where patients were registered. The social vulnerability index was developed by Haggerty et al. [45,50] and is based on four indicators: sources of social support, perception of one's financial situation, level of education completed and language spoken at home. Patients were categorized as having 1) no, 2) low or 3) high social vulnerability [45,50]. As presented above, all patients who used the GAP were asked to select the main professional or service they were referred to by the GAP for the primary reason for call. The variable was categorized into seven possible orientations: 1) receive a medical appointment with a family physician, 2) reference to another health professional, 3) reference to a community pharmacist, 4) reference to the emergency room, 5) receive a medical appointment with a nurse, 6) other reference (e.g., Info-Health, local community health center, community organization, other service) or 7) receive only information or advice from the GAP. LHTs are labelled 1, 2 and 3.

**Need factors** included self-rated mental and physical health (poor/fair; good/very good/excellent), and frequency of emergency room visits in the last 12 months. The frequency of use of the emergency room in the last 12 months was categorized as follows: no use, once, and twice or more.

## Analytical strategy

To characterize unattached patient populations, descriptive statistics of the sample and GAP orientations were performed. To assess patients' predisposing, enabling and need factors associated with perceived unmet needs (objective 1), binomial bivariate and multivariable logistic regressions were conducted. To contrast characteristics by GAP service received (objective 2), the multivariable regression models were stratified according to GAP orientation (models 1–7). All independent variables were included in multivariable models. Adjusted odds ratios (AOR) are presented with their 95% confidence intervals (CI). Statistical analyses were carried out using SPSS V29.0 (SPSS Inc., Chicago, IL, USA).

## Results

The characteristics of unattached patients who used the GAP are presented in Table 2. Participants were mostly female (60.02%), middle aged (35–69 years old), born in Canada (79.36%) and had no vulnerability (68.93%). More than half of the sample received a medical appointment with a family physician (56.00%). Most GAP users were in good physical (61.30%) and mental health (70.05%) and had not visited the emergency room in the last 12 months (67.29%).

**Table 2. Descriptive statistics of unattached patients who used GAP services in Montérégie region (n = 20 282).**

|  | N (%) |
|---|---|
| *Predisposing factors* | |
| Gender | |
| Female | 12,173 (60.02) |
| Male | 6,948 (34.26) |
| Missing | 1,161 (5.72) |
| Age | |
| 18-34 | 2,085 (10.28) |
| 35-54 | 6,430 (31.70) |
| 55-69 | 6,852 (33.78) |
| ≥70 | 3,883 (19.15) |
| Missing | 1,032 (5.09) |
| Born in Canada | |
| Yes | 16,096 (79.36) |
| No | 3,094 (15.26) |
| Missing | 1,092 (5.38) |
| *Enabling factors* | |
| Social vulnerability index | |
| No vulnerability | 13,981 (68.93) |
| Low vulnerability | 3,708 (18.28) |
| High vulnerability | 1,001 (4.94) |
| Missing | 1,592 (7.85) |
| GAP orientation | |
| Appointment with a family physician | 11,357 (56.00) |
| Reference to another health professional | 645 (3.18) |
| Reference to a community pharmacist | 541 (2.66) |
| Emergency room | 1,016 (5.00) |
| Appointment with a nurse | 1,043 (5.14) |
| Other reference | 3,644 (17.96) |
| Received information or advice only | 1,623 (8.02) |
| Missing | 413 (2.04) |
| LHT | |
| 1 | 4,720 (23.27) |
| 2 | 8,699 (42.89) |
| 3 | 6,863 (33.84) |
| Missing | 0 |
| *Need factors* | |
| Self-rated physical health | |
| Fair/poor | 5,325 (26.26) |
| Excellent/very good/good | 12,433 (61.30) |
| Missing | 2,524 (12.44) |
| Self-rated mental health | |
| Fair/poor | 3,619 (17.84) |
| Excellent/very good/good | 14,207 (70.05) |
| Missing | 2,456 (12.11) |

*(Continued)*

**Table 2.** (Continued)

| | N (%) |
|---|---|
| Frequency of emergency room visits | |
| No use | 13,648 (67.29) |
| One time | 3,120 (15.38) |
| Two times or more | 2,220 (10.95) |
| Missing | 1,294 (6.38) |

GAP: Primary Care Access Point for Unattached Patients (*Guichet d'accès à la première ligne*); LHT: Local Health Territory

The proportion of GAP orientations and related unmet healthcare needs for the main reason for call are presented in Fig 2. Perceived unmet needs varied widely from 17.90% for patients who received a medical appointment with a family physician to 71.83% for those who received information or advice only from the GAP.

Bivariate and multivariable regression models assessing the associations between predisposing, enabling and need factors and unmet healthcare needs are presented in Table 3. A total of 14,936 individuals had complete data for the variables under study and were included in multivariable analyses. The adjusted models showed that younger adults (<70 years old) were more likely to report unmet needs as were those with poor/fair self-rated physical and mental health. Compared to those who received a medical appointment with a family physician, patients who were oriented to a community pharmacist (AOR: 2.44, CI: 1.94–3.08) or to the emergency room (AOR: 2.80, CI: 2.38–3.30) were two times more likely to report unmet healthcare needs. Respondents oriented to other resources were six times more likely to report unmet needs (AOR: 6.70, CI: 6.07–7.40), while those who received information or advice only from the GAP were 11 times (AOR: 11.49, CI: 9.99–13.22) more likely. Receiving GAP services in LHT 3 compared to LHT 1 was associated with an increased likelihood of reporting unmet healthcare needs (AOR: 1.12, CI: 1.00–1.24). Finally, reporting at least one visit (one visit: AOR: 1.17, CI: 1.05–1.30; two and more visits: AOR: 1.25, CI: 1.11–1.42) to the emergency room in the last 12 months was associated with unmet needs compared to no visits.

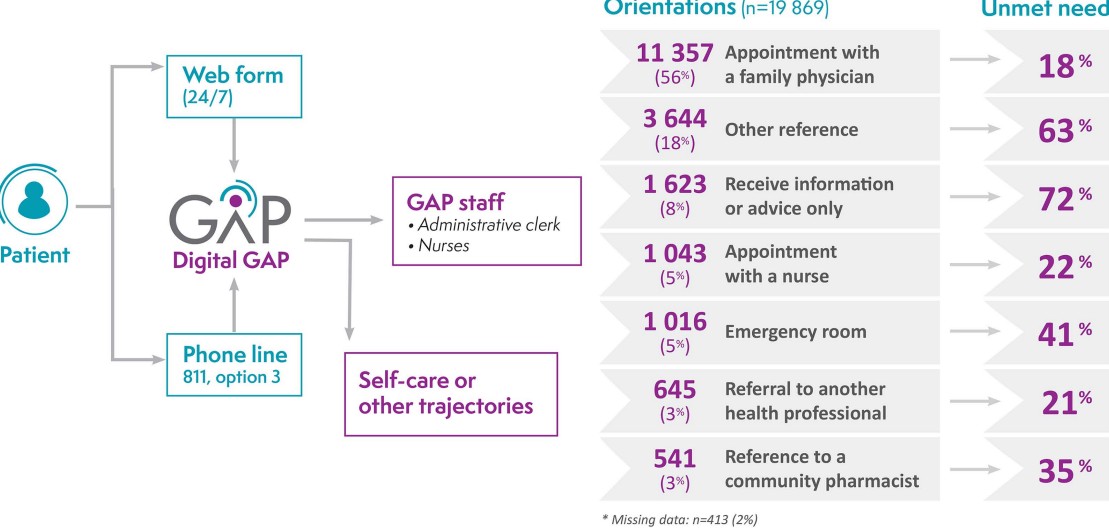

**Fig 2. Description of GAP orientations and related unmet healthcare needs.**

**Table 3. Bivariate and multivariable associations with unmet needs.**

| | Unmet needs<br>N = 5,989 | Met needs<br>N = 12,471 | Unadjusted<br>odds ratio | Adjusted odds ratio<br>N = 14,936 |
|---|---|---|---|---|
| *Predisposing factors* | | | | |
| Gender | | | | |
| Female | 3,646 (63.08) | 7,759 (63.83) | 0.97 (0.91-1.03) | 0.99 (0.92-1.08) |
| Male | 2,134 (36.92) | 4,396 (36.17) | Ref | Ref |
| Age | | | | |
| 18–34 years | 839 (14.38) | 1,122 (9.18) | **2.39 (2.12-2.68)** | **1.96 (1.68-2.29)** |
| 35–54 years | 2,282 (39.11) | 3,727 (30.51) | **1.95 (1.78-2.14)** | **1.82 (1.61-2.05)** |
| 55–69 years | 1,848 (31.67) | 4,604 (37.69) | **1.28 (1.17-1.41)** | **1.29 (1.14-1.45)** |
| ≥70 years | 866 (14.84) | 2,764 (22.62) | Ref | Ref |
| Born in Canada | | | | |
| Yes | 4,724 (81.10) | 10,370 (85.38) | Ref | Ref |
| No | 1,101 (18.90) | 1,776 (14.62) | **1.36 (1.25-1.48)** | 0.99 (0.89-1.11) |
| *Enabling factors* | | | | |
| Social vulnerability index | | | | |
| No vulnerability | 4,132 (72.88) | 9,000 (75.82) | Ref | Ref |
| Low vulnerability | 1,163 (20.51) | 2,296 (19.34) | **1.10 (1.02-1.20)** | 0.91 (0.82-1.00) |
| High vulnerability | 375 (6.61) | 574 (4.84) | **1.42 (1.24-1.63)** | 0.96 (0.80-1.14) |
| GAP orientation | | | | |
| Appointment with a family physician | 1,921 (32.11) | 8,811 (70.71) | Ref | Ref |
| Reference to another health professional | 105 (1.76) | 390 (3.13) | 1.24 (0.99-1.54) | 1.13 (0.88-1.45) |
| Community pharmacist | 171 (2.86) | 324 (2.60) | **2.42 (2.00-2.93)** | **2.44 (1.94-3.08)** |
| Emergency room | 383 (6.40) | 550 (4.41) | **3.19 (2.78-3.67)** | **2.80 (2.38-3.30)** |
| Appointment with a nurse | 207 (3.46) | 730 (5.86) | **1.30 (1.11-1.53)** | 1.16 (0.96-1.40) |
| Other reference (community organization, local health service center, Info-Health, other) | 2,040 (34.10) | 1,202 (9.65) | **7.78 (7.14-8.49)** | **6.70 (6.07-7.40)** |
| Receive information or advice only | 1,155 (19.31) | 453 (3.64) | **11.69 (10.38-13.18)** | **11.49 (9.99-13.22)** |
| LHT | | | | |
| LHT 1 | 1,239 (20.69) | 2,980 (23.90) | Ref | Ref |
| LHT 2 | 2,726 (45.52) | 5,217 (41.83) | **1.26 (1.16-1.36)** | 1.11 (0.99-1.23) |
| LHT 3 | 2,024 (33.79) | 4,274 (34.27) | **1.14 (1.05-1.24)** | **1.12 (1.00-1.24)** |
| *Need factors* | | | | |
| Self-rated physical health | | | | |
| Fair/poor | 1,953 (35.68) | 3,032 (27.08) | **1.49 (1.39-1.60)** | **1.53 (1.40-1.68)** |
| Excellent/very good/good | 3,521 (64.32) | 8,163 (72.92) | Ref | Ref |
| Self-rated mental health | | | | |
| Fair/poor | 1,386 (25.25) | 1,989 (17.69) | **1.57 (1.45-1.70)** | **1.28 (1.16-1.42)** |
| Excellent/very good/good | 4,104 (74.75) | 9,253 (82.31) | Ref | Ref |
| Frequency of emergency room visits | | | | |
| No use | 3,474 (66.21) | 9,045 (74.72) | Ref | Ref |
| One time | 993 (18.92) | 1,856 (15.33) | **1.39 (1.28-1.52)** | **1.17 (1.05-1.30)** |
| Two times or more | 780 (14.87) | 1,204 (9.95) | **1.69 (1.53-1.86)** | **1.25 (1.11-1.42)** |

GAP: Primary Care Access Point for Unattached Patients (*Guichet d'accès à la première ligne*); LHT: Local Health Territory

Significant results (p < 0.05) are in bold

Multivariable models were adjusted for all covariables included in Table 3.

The predisposing, enabling and need factors associated with unmet healthcare needs according to GAP orientations are presented in Table 4. Stratified analyses showed that being younger was associated with unmet needs for most orientations except reference to another health professional and appointment with a nurse. Being born outside of Canada was associated with a decreased likelihood of reporting unmet needs among those referred to other services (AOR: 0.68, CI: 0.54–0.84). Compared to no vulnerability, having low social vulnerability was associated with a decreased likelihood of reporting unmet needs in those who were oriented to other services (AOR: 0.72, CI: 0.58–0.90). Compared to patients from LHT 1, those in LHT 2 and 3 were more likely to report unmet needs when they received a medical appointment with a physician, although the association was only observed for LHT 2 in those who were oriented to other resources (AOR: 1.33, CI: 1.07–1.65). In contrast, among those who were oriented to the emergency room and who received information or advice only from the GAP, being from LHT 2 compared to LHT 1 was associated with a decreased likelihood of reporting unmet needs. Reporting poor or fair physical and mental health was associated with an increased likelihood of reporting unmet needs in patients who received a medical appointment with a family physician or nurse. However, among patients who received information or advice, only those with poor or fair mental health reported increased unmet needs. Using the emergency room in the last 12 months was associated with unmet needs in patients who received a medical appointment with a family physician, were referred to a community pharmacist, were oriented to other resources and who only received information or advice.

## Discussion

GAPs were implemented to help unattached patients navigate and access primary care services outside of the emergency room, in response to increasingly long wait times, while waiting to be attached through the centralized waiting lists. To our knowledge, this study is the first to document the factors associated with perceived unmet healthcare needs after receiving GAP service for the main reason for call and assess whether those factors vary according to GAP orientation. Using the Andersen framework [52], the study contributes to the limited literature on unmet healthcare needs among unattached patients. This is also the first study to evaluate GAP orientations from the patient perspective. Given that the GAP was implemented province-wide and is available to more than 1.6 million inhabitants, identifying the characteristics of unattached patients who are at higher risk of reporting unmet needs is fundamental to assess whether GAP's joint strategy–ensuring appropriateness of healthcare resources and optimal use of the expertise of health professionals other than family physicians–meets patient needs. Our study will also provide insights into GAP orientations that may be further developed or improved to meet more effectively the needs of unattached patients.

Findings suggest that most factors associated with perceived unmet healthcare needs are consistent with those found in other studies, such as being younger and having poor physical and mental health [13–15,24,25,27]. However, contrary to what has been reported across the literature [13,14,23,27,28,31,56], being a woman was not associated with reporting unmet healthcare needs. Previous studies have highlighted the potential role of social determinants (marital status, income, working status, education) in the association between gender, health and unmet healthcare needs [23,56]. However, previous studies did not exclusively include unattached patients. Further, in previous studies, the presence of unmet healthcare needs was assessed in reference to all healthcare needs [13,14,24–27], whereas in our study, unmet needs were assessed in relation to GAP services received for the main reason for call (e.g., medical appointment, reference to a professional, etc.).

Unmet needs reported by patients differed across the three LHTs. In fact, patients in LHT 3 were more likely to report unmet needs compared to those in LHT 1. Stratified analyses further showed that the increased unmet healthcare needs in patients from LHT 2 and 3 (compared to LHT 1) was only observed in individuals who received a medical appointment with a family physician and those who were referred to other resources. As shown in Table 1, LTH 1 has the lowest proportion of unattached patients. Comparisons of respondent's socio-demographic, economic and clinical factors amongst the three LHT showed that those from LTH 1 were younger (compared to LTH 2 and 3) and Canadian-born (compared to

**Table 4. Multivariable binomial logistic model (unmet needs, yes/no) stratified by GAP orientation.**

| | Appointment with a physician N=9,077 | Reference to another health professional N=406 | Community pharmacist N=352 | Emergency room N=815 | Appointment with a nurse N=755 | Other reference (community organization, local health service center, Info-Health, other) N=2,404 | Receive information only from the GAP N=1,178 |
|---|---|---|---|---|---|---|---|
| *Predisposing factors* | | | | | | | |
| Gender | | | | | | | |
| Female | 1.04 (0.93-1.17) | 1.43 (0.87-2.34) | 0.93 (0.56-1.52) | 0.79 (0.59-1.06) | 0.91 (0.62-1.35) | 1.03 (0.87-1.23) | 0.82 (0.62-1.08) |
| Male | Ref | Ref | Ref | Ref | Ref | Ref | Ref |
| Age | | | | | | | |
| 18–34 years | **2.11 (1.70-2.61)** | 1.19 (0.44-3.22) | **2.97 (1.23-7.18)** | **2.96 (1.66-5.30)** | 1.38 (0.69-2.75) | **1.48 (1.08-2.02)** | **2.33 (1.44-3.78)** |
| 35–54 years | **1.61 (1.36-1.91)** | 1.50 (0.68-3.33) | 1.71 (0.84-3.48) | **2.66 (1.70-4.19)** | 1.21 (0.65-2.24) | **2.17 (1.68-2.80)** | **2.19 (1.47-3.25)** |
| 55–69 years | 1.17 (0.99-1.38) | 1.24 (0.56-2.73) | 1.06 (0.55-2.03) | **1.65 (1.06-2.55)** | 1.05 (0.56-1.97) | **1.50 (1.17-1.94)** | 1.40 (0.96-2.05) |
| ≥70 years | Ref | Ref | Ref | Ref | Ref | Ref | Ref |
| Born in Canada | | | | | | | |
| Yes | Ref | Ref | Ref | Ref | Ref | Ref | Ref |
| No | 1.09 (0.94-1.28) | 1.46 (0.78-2.73) | 0.99 (0.52-1.91) | 0.87 (0.60-1.27) | 1.53 (0.98-2.38) | **0.68 (0.54-0.84)** | 1.01 (0.71-1.42) |
| *Enabling factors* | | | | | | | |
| Social vulnerability index | | | | | | | |
| No vulnerability | Ref | Ref | Ref | Ref | Ref | Ref | Ref |
| Low vulnerability | 0.98 (0.85-1.13) | 0.75 (0.38-1.48) | 0.63 (0.32-1.26) | 0.77 (0.54-1.10) | 1.31 (0.84-2.03) | **0.72 (0.58-0.90)** | 1.05 (0.76-1.46) |
| High vulnerability | 0.98 (0.76-1.25) | 1.48 (0.65-3.35) | 0.51 (0.12-2.17) | 1.30 (0.73-2.32) | 1.21 (0.63-2.33) | 0.82 (0.57-1.18) | 0.78 (0.41-1.47) |
| LHT | | | | | | | |
| LHT 1 | Ref | Ref | Ref | Ref | Ref | Ref | Ref |
| LHT 2 | **1.22 (1.06-1.40)** | 1.20 (0.58-2.49) | 1.40 (0.63-3.10) | **0.66 (0.46-0.94)** | 0.88 (0.50-1.53) | **1.33 (1.07-1.65)** | **0.58 (0.40-0.86)** |
| LHT 3 | **1.16 (1.00-1.34)** | 0.90 (0.42-1.93) | 1.55 (0.66-3.65) | 0.92 (0.63-1.35) | 1.02 (0.58-1.79) | 1.22 (0.97-1.52) | 0.73 (0.47-1.12) |
| *Need factors* | | | | | | | |
| Self-rated physical health | | | | | | | |
| Fair/poor | **1.93 (1.71-2.18)** | 1.27 (0.72-2.24) | 0.82 (0.46-1.47) | 1.08 (0.79-1.47) | **1.72 (1.13-2.60)** | 1.16 (0.95-1.42) | 1.02 (0.74-1.41) |
| Excellent/very good/good | Ref | Ref | Ref | Ref | Ref | Ref | Ref |
| Self-rated mental health | | | | | | | |
| Fair/poor | **1.25 (1.09-1.44)** | 1.16 (0.60-2.24) | 1.67 (0.80-3.47) | 1.18 (0.81-1.71) | **1.59 (1.01-2.50)** | 1.22 (0.97-1.53) | **1.55 (1.07-2.24)** |
| Excellent/very good/good | Ref | Ref | Ref | Ref | Ref | Ref | Ref |
| Frequency of emergency room visits | | | | | | | |
| No use | Ref | Ref | Ref | | Ref | Ref | Ref |
| One time | **1.28 (1.10-1.49)** | 0.82 (0.38-1.75) | 1.18 (0.60-2.33) | | 0.84 (0.49-1.45) | 1.19 (0.95-1.48) | 1.31 (0.92-1.85) |
| Two times or more | **1.32 (1.10-1.58)** | 1.49 (0.74-2.97) | **2.46 (1.04-5.85)** | | 1.61 (0.96-2.72) | **1.37 (1.04-1.80)** | **1.63 (1.04-2.55)** |

GAP: Primary Care Access Point for Unattached Patients (*Guichet d'accès à la première ligne*); LHT: Local Health Territory

Significant results (p<0.05) are in bold

Multivariable models were adjusted for all covariables included in Table 4 with the exception of the emergency room model, which was adjusted for all covariables except frequency of emergency room visits.

LTH 3). Other factors not documented could also have potentially influenced the results such as retention of GAP personnel and labor shortage. The findings could also be partly explained by external factors. On June 1, 2024, the 2-year agreement for GAP appointments between family physicians and the Ministry of Health and Social Services ended, leading to a large decrease in the number of appointments with a family physician available for GAP patients [57]. The survey was launched in LHT 1 before the termination of the agreement, whereas in LHT 2 and 3, it was sent a few days before or after the termination. This could have highly influenced reported unmet needs considering that, during this period, patients were oriented to other professionals or services (including emergency room and private clinics) or were put on waiting lists. This highlights the importance of incentives for family physicians for this type of organizational innovation. A new agreement was signed in mid-June 2024, but the number of available appointments was lower than usual for several weeks after it was signed [58]. During this period of uncertainty, the GAPs relied on other professionals and orientations to meet patient needs. However, the results of a qualitative study with key actors and healthcare professionals working in four GAPs and at the provincial level showed that these other orientations are underused and require further development and appropriate funding to reach the level of services offered by family physicians [59]. This is in line with primary care transformations around the world, which are moving towards more interprofessional practice and calling on professionals other than physicians to contribute to team-based care [60–63].

Our results suggest that receiving a medical appointment seems to meet the needs of unattached patients in most cases (82%), whereas unmet needs reached up to 72% for other orientations. One possible explanation why individual oriented to other professional or service were more likely to report unmet needs is that patients might expect to receive an appointment with a family physician when they use the GAP. These patients might have experienced access barriers for many years and seen the implementation of the GAP as an opportunity to seek care. In fact, 62% of the sample had not had a family physician for at least 3 years and 30% for at least 5 years. It was previously shown that patients who expected to see a physician and were seen by a nurse reported worse patient experiences [64]. From an interprofessional perspective, a cultural shift is also needed within the population by improving communication with patients and enhancing their understanding of the scope and role of each professional. The Quebec Ministry of Health and Social Services's main message 'The right service, by the right person, at the right time!' [65] should also be further promoted, for example through flyers in waiting rooms, social media, television and radio campaigns, as well as direct discussion with patients during consultations.

The orientation 'other resources' included services requested by patients and not covered by the GAP but available to attached patients (e.g., medical check-ups, some preventive tests). This result might be indicative of the major limitation inherent in the GAP design, which is intended to only address health needs that are prompt and acute. While the GAP improves access to primary care services for unattached patients, its design falls short of meeting the core components of primary care, particularly comprehensiveness and continuity of care [66]. Although interdisciplinary team care is available within the GAP for some patients with chronic diseases, services provided to GAP patients do not cover the same services available for attached patients in terms of preventive, comprehensive and continuity of care (e.g., seeing the same physician/NP, accessing medical check-ups, consultations with other professional in a public clinic and accessing some preventive tests). Interviews conducted with various GAP staff members and health professionals revealed that patients were often seeking a medical check-up or preventive tests but were unable to receive them given that GAP services are meant to be for acute health problems rather than health promotion and preventive medicine [59]. Studies have also shown that unattached patients are less likely to receive preventive and comprehensive care [9,10,67], leading to worse health outcomes [12,68].

The main model of primary care clinics in Quebec is the Family Medicine Group, to which about 65% of the population is attached. These clinics typically include family physicians, nurse practitioners, clinical nurses, social workers and other professionals based on the needs of the clinic (i.e., physiotherapist, pharmacist, psychologist). A report produced by an expert committee following a provincial tour of all GAPs in Quebec also highlighted the limited services provided to

GAP patients and recommended offering the same services as those available to attached patients [69]. Based on our findings, we recommend eliminating the distinction between patient categories with unequal access to care and, consequently, phasing out a 'patching' structure like the GAP. The Quebec's Ministry of Health and Social Services is currently developing its first primary care policy, with one of the key objectives being to have 100% of the population registered with a primary care clinic by the summer of 2026 [70]. In this context, it is likely that the GAP will eventually be phased out or assigned a transitional role.

### Strengths and limitations

Despite the large sample size of the study, stratified analyses may have lacked statistical power for some orientations (i.e., appointment with other health professionals, community pharmacists). The study sample, representing only 7% of all unattached patients registered on the centralized waiting list in Montérégie, was limited to people with an email address, which may have led to the overrepresentation of individuals with high digital literacy. Unattached patients who are not registered on the centralized waiting list were also not captured in our study. Therefore, vulnerable individuals (i.e., older adults, individuals living in poverty or with low digital literacy) may be underrepresented in the study sample. To reach these groups, additional strategies should be considered in future studies, such as postal questionnaires and phone administration [71,72]. However, these data collection methods are typically more expensive. Further, data on the characteristics of unattached patients at the regional level are not available, and this population is changing daily with new patients being registered, while others are removed after being assigned a GP. This limited our ability to compare the characteristics of the available sample and the target population and apply sample weighting to address the potential bias. However, public data available for the entire population of Montérégie (attached and unattached individuals) showed that the residents are more economically advantaged (higher employment rate, average wage and disposable income/capita) and the region has more immigration compared to the province of Quebec [73]. Unmet needs were only measured for the GAP service received for the main reason for call, while not considering the other health needs discussed during the call. This limitation could be even greater for complex cases and multimorbidity patients.

Despite these limitations, we documented unmet healthcare needs in primary care, a setting for which the literature on unmet needs remains scant, especially for unattached patients. This study relied on patient-reported experience measures (PREM), which provides a unique perspective by documenting unmet needs through the direct experience of patients, the actual users of primary healthcare services. This is also the first study to document users' unmet needs related to the GAP services received, a necessary step to provide decision-makers with evidence-based recommendations to improve patients' experiences using the GAP. The next step will be to use qualitative data from unattached patients who have used the GAP to dig deeper into the reasons behind patients' unmet healthcare needs, leading to more accurate strategies for reducing them.

### Conclusion

This study serves as a first step in deepening our understanding of how to better plan primary care services and improve unattached patients' experiences using the GAP. While some characteristics were associated with unmet needs across orientations, others were associated with specific orientations. This study showed that patients receiving a medical appointment with a family physician had the lowest unmet needs compared to other orientations. The next step will be to better understand the reasons why patients' needs are unmet.

### Acknowledgments

The authors wish to thank Lisa Starr for scientific and linguistic editing. We are also deeply grateful to the patients who participated in this study by completing the questionnaire.

## Author contributions

**Conceptualization:** Mylaine Breton, Catherine Lamoureux-Lamarche, Véronique Deslauriers.

**Formal analysis:** Catherine Lamoureux-Lamarche.

**Funding acquisition:** Mylaine Breton, Maude Laberge.

**Supervision:** Mylaine Breton.

**Writing – original draft:** Mylaine Breton, Catherine Lamoureux-Lamarche, Véronique Deslauriers.

**Writing – review & editing:** Djamal Berbiche, Maude Laberge, Annie Talbot, Aude Motulsky, Marie-Pascale Pomey, Isabelle Gaboury.

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
