## [Decision Letter · Decision Letter 0]

11 Sep 2025

Dear Dr. Breton,

Thank you for submitting your manuscript to PLOS ONE. After careful consideration, we feel that it has merit but does not fully meet PLOS ONE’s publication criteria as it currently stands. Therefore, we invite you to submit a revised version of the manuscript that addresses the points raised during the review process.

The reviewers were positive about the manuscript but described some areas that would benefit from improvements. In particular, providing contextual information on the setting and the measurement tools used. Please see the detail of the comments below.

We look forward to receiving your revised manuscript.

Kind regards,

Aliah Faisal Shaheen

Academic Editor

PLOS ONE

Journal Requirements:

Journal Requirements:

“I have read the journal's policy and the authors of this manuscript have the following competing interests: AT mentioned that a family member is working for a pharmaceutical company. She also received an honorarium as consultant to evaluate the GAP implementation in Quebec from MSSS.”

5.If the reviewer comments include a recommendation to cite specific previously published works, please review and evaluate these publications to determine whether they are relevant and should be cited. There is no requirement to cite these works unless the editor has indicated otherwise.

Reviewers' comments:

Reviewer's Responses to Questions

**Comments to the Author**

1. Is the manuscript technically sound, and do the data support the conclusions?

Reviewer #1: Yes

Reviewer #2: Yes

2. Has the statistical analysis been performed appropriately and rigorously?

Reviewer #1: Yes

Reviewer #2: Yes

3. Have the authors made all data underlying the findings in their manuscript fully available?

Reviewer #1: Yes

Reviewer #2: No

4. Is the manuscript presented in an intelligible fashion and written in standard English?

Reviewer #1: Yes

Reviewer #2: Yes

Reviewer #1: Strengths of the Study

- Use of a very large, multi-site patient sample enhances generalizability of findings.

- Patient-centered outcomes are measured using validated PREM tools, directly reflecting patient experiences.

- Application of the Andersen behavioral model allows nuanced analysis of predisposing, enabling, and need factors.

- Stratified regression models by GAP orientation provide granular insights into modality-specific issues.

- The authors contextualize findings within ongoing healthcare reforms and recent changes to physician incentive structures, revealing important external influences.

Areas for Improvement and Additional Considerations

- Response Rate and Representativeness

The response rate in this study warrants critical attention in evaluating the generalizability of its findings. As recruitment relied on email invitations, there is a risk of underrepresenting vulnerable groups (elderly, low-income, or digital literacy-limited populations). Please provide further discussion of this limitation, and—if feasible—consider applying or discussing sample weighting strategies to address non-response and improve credibility. According to the manuscript: A total of 212,546 patients with valid email addresses were invited to participate in the survey from a pool of approximately 279,000 unattached patients registered across three local health territories (LHTs) in Quebec. Of these, 41,384 individuals responded, yielding an overall response rate of approximately 19% among those invited. Among respondents, 20,282 individuals both responded on their own behalf and had used the GAP service—forming the analytic sample. This implies that the analyzed data represents only about 9.5% of invited individuals, and approximately 7.3% of the total unattached patient population in the target regions.

- Limitations of the GAP Model for Core Primary Care Functions

The current GAP model is optimized for prompt, acute care. I recommend a more explicit discussion of the structural gaps in preventive, chronic, and continuous care for unattached patients, and—if possible—suggest strategies or policy directions to address these limitations.

- Definition of Unmet Need

As the measurement of unmet need was specifically focused on the GAP encounter for the main reported reason, emphasize this scope in both Results and Discussion. Also, mention as a limitation that multiple, concurrent unmet needs may exist, particularly among complex patients.

- Patient–Provider Education in Team-Based Care

Since unmet need was substantially higher for non-physician provider orientations, the discussion should address not only patient expectations but also opportunities for better communication, patient education, and acceptance of diverse provider roles within interprofessional primary care.

- Conclusion and Recommendation -

While the study offers valuable insights into the characteristics and experiences of GAP users who responded to the survey, its findings should not be generalized to all unattached patients in Quebec without caution. Future studies would benefit from: Comparative analyses of respondent vs. nonrespondent characteristics, application of population weights or statistical correction methods, and use of alternative survey modalities (e.g., phone, mail) to reach digitally underserved populations.

Overall, this is an important study that substantially contributes to our understanding of access models for unattached patients and the design of team-based primary care systems. The large-scale, real-world data, careful methodology, and clear policy relevance are strengths.

Provided that the authors address the above points—particularly around issues of representativeness, the definition of unmet need, and strategies to build acceptance and effectiveness of team-based care—I find this manuscript suitable for publication in PLOS ONE.

Reviewer #2: Dear Authors,

It is very interesting and valuable topic of research as the objectives of this study are to" 1) document the factors associated with unmet healthcare needs after receiving a GAP service and 2) assess if these factors vary according to GAP service received" So goals of research confirm the importance of the research as well as they are clearly specified.

Introduction contains not only the description of purposes of the research but also the importance of primary health and unmet needs are well explained. Also factors, which are responsible for unmet needs are mentioned and they apply to Canada. So, it would be worth to make also some literature research of healthcare system, which are similar type of Canada in purpose to see also what are the factors of unmet needs and then to consider them also if they are different. So, readers should be aware if they are the same etc..

Methodology is well explained and especially the questionnaire.

Results and discussion are well presented. However the practical and theoretical implications should be expended.

**Do you want your identity to be public for this peer review?** For information about this choice, including consent withdrawal, please see our Privacy Policy

Reviewer #1: No

Reviewer #2: No

---

## [Author Response · Author response to Decision Letter 1]

7 Oct 2025

The respond to reviewers has been uploaded as an attached file.

---

## [Decision Letter · Decision Letter 1]

25 Nov 2025

Dear Dr. Breton,

**The reviewer has made suggestions on the interpretation and analysis of the results in light of the limitations. Please address these further comments.**

We look forward to receiving your revised manuscript.

Kind regards,

Aliah Faisal Shaheen

Academic Editor

PLOS ONE

**Journal Requirements:**

Reviewers' comments:

Reviewer's Responses to Questions

**Comments to the Author**

Reviewer #1: All comments have been addressed

2. Is the manuscript technically sound, and do the data support the conclusions?

Reviewer #1: Yes

3. Has the statistical analysis been performed appropriately and rigorously?

Reviewer #1: Yes

4. Have the authors made all data underlying the findings in their manuscript fully available?

Reviewer #1: Yes

5. Is the manuscript presented in an intelligible fashion and written in standard English?

Reviewer #1: Yes

**Reviewer #1:** Thank you for your detailed response addressing the concerns raised regarding the response rate and representativeness. I appreciate the authors’ clear acknowledgment of the limitations inherent in using an email-based recruitment strategy, as well as the expanded discussion provided in the revised manuscript. Your explanation regarding the lack of available population characteristics for unattached patients across the three LHTs—and the high turnover within the CWL—offers a reasonable justification for why weighting strategies were not feasible.

That said, I encourage the authors to further strengthen the manuscript in two areas:

1. Interpretation of the potential impact on study findings.

While the limitations are acknowledged, the revised version does not sufficiently discuss how low representativeness may affect the direction or magnitude of key results. Because the analytic sample represents only about 7% of the unattached population, it would be helpful to explicitly address how overrepresentation of individuals with higher digital access might influence estimates of unmet healthcare needs or the distribution of GAP orientations.

2. Consideration of alternative post-hoc adjustment approaches.

Even if full weighting was not possible, it may still be beneficial to briefly note whether any partial adjustment strategies were considered—such as LHT-level calibration, or comparisons with publicly available demographic data. Stating that such options were evaluated but deemed infeasible would improve transparency in methodological decision-making.

Your revisions are appreciated and address the core of the concern; however, a modest additional elaboration on these points would better contextualize the implications of non-response and representativeness limitations for readers and policymakers who may rely on these findings.

Minor Comments

1. Figures and Tables

Please improve the readability of Figure 2 by using clearer labels or considering an alternative graphical format.

Ensure consistent formatting of statistically significant results across tables (e.g., bolding, decimal places).

Add the following footnote to Table 1 to define all abbreviations:

LHT: Local Health Territory

CWL: Centralized Waiting List for Unattached Patients

GAP: Primary Care Access Point for Unattached Patients

2. Clarity and Grammar

Minor edits are needed to improve clarity:

“an in-depth exploration the reasons” → “an in-depth exploration of the reasons”

Thank you again for your careful revisions.

**Do you want your identity to be public for this peer review?** For information about this choice, including consent withdrawal, please see our Privacy Policy

Reviewer #1: No

---

## [Author Response · Author response to Decision Letter 2]

11 Dec 2025

The response to reviewers is attached to the submission.

---

## [Decision Letter · Decision Letter 2]

5 Jan 2026

Dear Dr. Breton,

**The remaining recommendations by the reviewer are very minor, please ensure that they are implemented in the next version to speed up the publication process.**

plosone@plos.org . A letter that responds to each point raised by the academic editor and reviewer(s). You should upload this letter as a separate file labeled 'Response to Reviewers'.A marked-up copy of your manuscript that highlights changes made to the original version. You should upload this as a separate file labeled 'Revised Manuscript with Track Changes'.An unmarked version of your revised paper without tracked changes. You should upload this as a separate file labeled 'Manuscript'.

We look forward to receiving your revised manuscript.

Kind regards,

Aliah Faisal Shaheen

Academic Editor

PLOS One

**Journal Requirements:**

Reviewers' comments:

Reviewer's Responses to Questions

**Comments to the Author**

Reviewer #1: All comments have been addressed

2. Is the manuscript technically sound, and do the data support the conclusions?

Reviewer #1: Yes

3. Has the statistical analysis been performed appropriately and rigorously?

Reviewer #1: Yes

4. Have the authors made all data underlying the findings in their manuscript fully available?

Reviewer #1: No

5. Is the manuscript presented in an intelligible fashion and written in standard English?

Reviewer #1: Yes

**Reviewer #1:** The revised manuscript has improved substantially since the previous submission and now presents a clear and well-structured analysis of unmet healthcare needs among unattached patients using Primary Care Access Points (GAP). The study addresses a relevant primary care issue, and the analytical approach is generally appropriate. Most of the major concerns raised in the previous review have been adequately addressed.

However, a few minor issues related to reporting clarity remain and should be corrected before final acceptance.

First, in Table 3, adjusted odds ratios (AORs) are presented, but the table footnote does not specify which covariates were included in the adjusted model. For transparency and ease of interpretation, the authors should clearly state in the footnote the variables adjusted for in the multivariable analysis.

Second, Table 4 raises the same issue as Table 3. Adjusted estimates are shown, yet the corresponding footnote does not describe the adjustment variables. Consistent reporting across tables is important, and Table 4 should include a footnote specifying the covariates used in the adjusted analysis, in parallel with Table 3.

Third, all tables should be self-explanatory, with clear definitions of abbreviations used. In particular, the abbreviation “GAP” should be defined in the footnote of each table where it appears, even if it has been defined elsewhere in the text.

Overall, the authors have responded appropriately to the previous reviewer comments, and the remaining issues are minor and editorial in nature. These can be addressed easily without additional analysis.

I recommend acceptance after minor revision, and further external peer review does not appear necessary.

**Do you want your identity to be public for this peer review?** For information about this choice, including consent withdrawal, please see our Privacy Policy

Reviewer #1: No

---

## [Author Response · Author response to Decision Letter 3]

8 Jan 2026

The response to reviewers was added as an attached file.

---

## [Editor Report · Decision Letter 3]

19 Jan 2026

Factors associated with unmet healthcare needs in patients using Primary Care Access Points for unattached patients in Quebec (Canada)

PONE-D-25-23656R3

Dear Dr. Breton,

We’re pleased to inform you that your manuscript has been judged scientifically suitable for publication and will be formally accepted for publication once it meets all outstanding technical requirements.

Kind regards,

Aliah Faisal Shaheen

Academic Editor

PLOS One
---

## [Editor Report · Acceptance letter]

PONE-D-25-23656R3

PLOS One

Dear Dr. Breton,

I'm pleased to inform you that your manuscript has been deemed suitable for publication in PLOS One. Congratulations! Your manuscript is now being handed over to our production team.

Kind regards,

on behalf of

Dr. Aliah Faisal Shaheen

Academic Editor

PLOS One